# OTLDA: A Geometry-Aware Optimal Transport Approach for Topic Modeling

**Viet Huynh, He Zhao, Dinh Phung**
Faculty of Information Technology
Monash University, Australia
`viet.huynh, ethan.zhao,dinh.phung@monash.edu`

## Abstract

We present an optimal transport framework for learning topics from textual data. While the celebrated Latent Dirichlet allocation (LDA) topic model and its variants have been applied to many disciplines, they mainly focus on word-occurrences and neglect to incorporate semantic regularities in language. Even though recent works have tried to exploit the semantic relationship between words to bridge this gap, they, however, these models which are usually extensions of LDA or Dirichlet Multinomial mixture (DMM) are tailored to deal effectively with either regular or short documents. The optimal transport distance provides an appealing tool to incorporate the geometry of word semantics into it. Moreover, recent developments on efficient computation of optimal transport distance also promote its application in topic modeling. In this paper we ground on optimal transport theory to naturally exploit the geometric structures of semantically related words in embedding spaces which leads to more interpretable learned topics. Comprehensive experiments illustrate that the proposed framework outperforms competitive approaches in terms of topic coherence on assorted text corpora which include both long and short documents. The representation of learned topic also leads to better accuracy on classification downstream tasks, which is considered as an extrinsic evaluation.

## 1 Introduction

Topic models such as Latent Dirichlet Allocation (LDA) [3] and its extensions have been successfully applied to various domains such as science publication, social science, and machine translation [4]. Fundamentally, topic models are probabilistic models that infer a set of latent topics from a corpus using word co-occurrences within each document. However, when there are a small number of documents or the corpus contains short documents, topic models will tend to infer poor quality topics from little evidence of co-occurrences. Moreover, infrequently occurring words in the corpus might be grouped into irrelevant topics although there are significant statistics of synonyms of those words in the corpus.

Several existing studies have targeted to incorporate synonyms or semantic relationship between words into topic models to improve the topic representations. The source of semantic regularities may come from thesauri or knowledge base [20, 25] or distributional similarity [25, 28, 19]. Distributional representation of word semantics, aka word embeddings, has been widely utilized to improve the performance as well as the robustness of learned topics. Much research focuses on adapting existing frameworks to integrate semantic relationships of words [19, 28, 1, 7], while other authors developed non-conjugate models for topic modeling with word embedding awareness and inference with using deep neural networks [8, 14].

Another notable strand developed recently is the optimal transport theory which has been employed in formulating loss functions to numerous machine learning problems, particularly in language modeling such as learning document distances [13, 11], generative models [17]. Optimal transport geometry was also applied to quantify the fit between the observed matrix and its reconstruction in the nonlinear dictionary learning [23, 22, 26]. When the observed matrix is the normalized word count matrix, the dictionary learning problem is closely related to topic modeling [9]. The optimal transport distance becomes appealing to machine learning problems due to its nature of integrating the geometry of the data space of the distributions. Recent developments on efficient computation of optimal transport distance [2, 5, 6] also promote its application in machine learning.

In this paper, we aim to apply optimal transport theory to semantically modeling document topics with geometry awareness of word embedding space. The benefit of using optimal transport distance is to naturally exploit the geometric structures of semantically related words in embedding spaces which leads to more interpretable learned topics. Under the convex geometric perspective of the Latent Dirichlet Allocation (LDA) or probabilistic latent semantic indexing (PLSI) [10, 27], our proposed framework is a generalized methodology of LDA/PLSI in which the loss of squared Euclidean distance between a document empirical distribution and its topic mixture is substituted by regularized optimal transport distance (aka Wasserstein distance). Our model also distinguishes from nonlinear dictionary learning models [23, 22, 26] in the aspect that we take document length into account which allows to effectively model heterogeneous corpora of long and short documents.

In summary, our contributions in this paper are: (i) we provide a novel representation which generalizes the geometric loss function of LDA/PLSI via discrete optimal transport framework. The framework naturally allows us to incorporate underlying geometry of word semantics in embedding spaces; (ii) our proposed formulation leads to an efficient learning algorithm using alternating optimization borrowed from discrete optimal transport optimization techniques; (iii) our proposed model achieves significantly better performance in the comparison with state-of-the-art topic models; (iv) last but not least, with a strong flavor of geometry and efficient optimization, our framework has implications to study richer classes of topic models such as with multilevel, hierarchical or temporal structures.

**Notations**. We denote a corpus of $N$ documents of vocabulary size $V$ by $D = \{d_i\}_{i=1}^N$. Each document contains $n_i$ (repeated) word counts and is represented as a normalized empirical distribution on the support of $V$ vocabulary: $d_i = \frac{1}{n_i} \sum_{v=1}^V n_{iv} \delta_{w_v}$, where $n_{vi}$ is the number of word $v$ in document $i$. We also denote the normalized word count of document $i$ as $\bar{\boldsymbol{n}}_i$, a (sparse) vector of $V$ dimensions. The collection of $K$ learned topics is denoted as $\boldsymbol{B} = \{\beta_k\}_{k=1}^K$ where $k$-th topic belongs to the simplex $\Sigma^{V-1}$ of $\mathbb{R}^V$. Regularized optimal transport distance between two distributions $p$ and $q$ is written as $\text{OT}_\gamma(p, q)$.

## 2 Related background

In this section, together with related work, we review key results on optimal transport, distance, and barycenter, as well as the geometric view of the notable LDA model that we are using in subsequent sections for developing our proposed framework. We also review literature related to our work.

### 2.1 Optimal transport distance and LDA geometry interpretation

Let $p$ and $q \in \mathcal{P}(X)$ be two discrete probability distributions on the arbitrary space $X \subseteq \mathbb{R}^n$ endowed with cost function $d$ between two points $x, y \in X$. Suppose that $p$ and $q$ share the fixed number of supports $V$ which means $p = \sum_{v=1}^V r_v \delta_{x_v}$ and $q = \sum_{v=1}^V c_v \delta_{x_v}$ where $\boldsymbol{u}, \boldsymbol{v} \in \Sigma^V$, the simplex of $\mathbb{R}^V$.

**Optimal transport distance** between $p$ and $q$ is defined as the optimization problem $\text{OT}(p, q) = \min_T \sum_{u,v} t_{uv} d_{uv}$, such that $\sum_v^V t_{uv} = r_u$ and $\sum_u^V t_{uv} = c_v$. Here, $d_{uv} = d(x_u, x_v)$ and $T$ is a $V \times V$ matrix called transportation plan in which $t_{uv}$ is an element at row $u$, column $v$. As computing the distance has the cubic time complexity, Cuturi [5] suggested using the entropic regularization, $H = -\sum_{uv} t_{uv} \ln t_{uv}$, to relax the problem and lead to fast computation.

**Optimal transport barycenter** is a notion of Fréchet mean of a set of discrete probability distributions $\{p_1, \ldots p_m\}$ which is defined as the minimizer of the following optimization problem

$\underset{q \in \mathcal{P}(X)}{\operatorname{argmin}} \sum_{i=1}^{m} \lambda_i \operatorname{OT}(q, p_i)$ where $\lambda_i > 0$ and $\sum_i \lambda_i = 1$. Similar to optimal transport distance, the problem of finding the barycenter needs a high complexity algorithm to compute. However, relaxing this problem with entropic regularization also leads to a smooth problem with efficient algorithms to solve [2, 6].

**Optimal transport (Wasserstein) dictionary learning** [23, 22] is an extension of non-negative matrix factorization in which the loss function $l_2$ of reconstruction error is substituted by Wasserstein distance. Let $D = \{d_i\}_{i=1}^{N}$ be normalized bag-of-word of documents where $d_i$ in $V$-dimensional simplex. Dictionary learning aims to learn to factorize $D$ into $K$ dictionary elements $\boldsymbol{B} = \{\beta_k\}_{k=1}^{K}$ of the same dimension $V$ and matrix mixture weights $\Lambda = \{\lambda_i\}_{i=1}^{N}$. The objective of learning is to solve problems of form $\min_{B, \Lambda} \mathcal{L}(d_i, \sum \lambda_{ik}\beta_k)$. When loss function $\mathcal{L}$ is squared Euclidean distance the problem becomes non-negative matrix factorization (NMF). Otherwise, if $\mathcal{L}$ is Kullback–Leibler divergence, it becomes PLSI [9]. In Wasserstein dictionary learning, authors dedicated to using Wasserstein (optimal transport) distance for loss function $\mathcal{L}$. Dictionary elements $\boldsymbol{B}$ can be interpreted as the topics while $\Lambda$ can be considered as document topic proportions.

**Convex geometry of topics** interpretation of LDA was introduced in [27, 3, 10] in which learning topics from documents in LDA model is equivalent to estimate the convex hull of the $K$ topics $\boldsymbol{\Omega} = \operatorname{Conv}(\boldsymbol{\omega}_1, \ldots, \boldsymbol{\omega}_K)$ from noisy observation of documents. Here $\boldsymbol{\omega}_k$ represents a topic to be learned in LDA. The authors surrogate the LDA's likelihood with the geometric loss function as $\min_{\boldsymbol{\Omega}} \sum_{i=1}^{N} n_i \min_{\boldsymbol{\omega}_i \in \boldsymbol{\Omega}} \|\boldsymbol{\omega}_i - \bar{\boldsymbol{n}}_i\|_2^2$, where $\bar{\boldsymbol{n}}_i$ is normalized word counts in document $i$. Based on this interpretation, we propose replacing squared $l_2$ loss with optimal transport distance to incorporate the underlying geometry of word embedding space.

## 2.2 Related work

Since the seminal work on learning distributional representation of word semantics using neural networks was introduced [15], there are amount of works attempts to exploit this information into topic models to improve interpretation of learned topics. There is previous research pursuing this goal by extending LDA models. These methods modify the priors [28, 24] or replace the Multinomial likelihood with a Gaussian [7] or a Von-Mises Fisher [1] likelihood to handle continuous observations of embeddings. Some other research has customized LDA by combining the original likelihood with the embedding likelihood [19].

Other recent research was developed non-conjugate models for topic modeling with word embedding awareness and inference with using deep neural networks [8, 14]. These methods construct the variational distributions as deep neural networks called inference network and optimize the evidence lower bound (ELBO) which is a lower bound for the divergence between the model and the variational distribution.

Yurochkin et. al. [27] have introduced the geometry interpretation for LDA and proposed an efficient algorithm for learning LDA. However, their method is designed for learning LDA which neglects word semantics. Our proposed framework can be viewed as an improvement of this work by naturally incorporating the word semantics into modeling.

Optimal transport distance has been used to learn topics from textual data in [22, 23, 26]. These models aim to learn topics and document topic proportions from the normalized bag-of-word matrix of documents by nonlinear dictionary learning. In addition, the work of [26] extends optimal transport dictionary learning in [23] by jointly learning a cost function between words in the vocabulary. These models are closely related to our proposed model, we elaborate on the differences and geometric view in Section 3.1.

## 3 Geometry-aware optimal transport approach to topic modeling

### 3.1 Documents, topics and problem formulation

The input for our model is similar to that of LDA which consists of $N$ documents in normalized bag-of-words representation. Each document $i$ has $n_i$ words in the list of vocabulary, and each word $v$ may occur more than once in the document and its occurrences denoted $n_{iv}$. The normalized

bag-of-word representation of each document $i$, $d_i$, can be represented as an empirical distribution. Similarly, the corpus including $N$ documents are represented as an empirical distribution $P$ on the space of (empirical) document distributions as follows:

$$d_i = \frac{1}{n_i} \sum_{v=1}^{V} n_{iv} \delta_{w_v} \qquad\qquad P = \frac{1}{\sum_i n_i} \sum_{i=1}^{N} n_i \delta_{d_i}.$$

The aim of the model is to learn topics from the corpus where topics are represented as a collection of $K$ distributions $\mathbf{B} = \{\beta_k\}_{k=1}^{K}$, each of them has the same support with documents, i.e. $\beta_k = \sum_v \omega_v \delta_{w_v}$. Collectively, we form a topic distribution $Q$ (on the space of topics) defined as

$$Q = \sum_{k=1}^{K} b_k \delta_{\beta_k}. \tag{1}$$

Our goal is to learn the topic distribution $Q$ that is close to the empirical distribution $P$. To this end, we use the *regularized optimal transport distance* as the divergence, bearing a similarity to LDA, except instead of viewing the generative process as an admixture view with squared $l_2$ loss function [27], our model directly minimizes the discrete optimal transport between P and Q, hence we term our model Optimal Transport based LDA, OTLDA. Topics and document topic proportions are the solutions to the following optimization problem

$$\min_{B,\mathbf{b},\boldsymbol{\pi}} \mathrm{OT}_\gamma(P,Q) = \min_{B,\mathbf{b},\boldsymbol{\pi}} \sum_{k=1}^{K} \sum_{i=1}^{N} \pi_{ik} \mathrm{OT}_\gamma(\beta_k, d_i) - \gamma H(\boldsymbol{\pi}), \tag{2}$$

such that $\sum_i \pi_{ik} = b_k$ and $\sum_k \pi_{ik} = \frac{n_i}{\sum_i n_i}$ where $\boldsymbol{\pi} = \{\pi_{ik}\}_{i=1,k=1}^{N,K}$. Here, $\mathrm{OT}_\gamma(\cdot \mid \cdot)$ again is the regularized optimal transport distance between document and topic and is used as a ground distance between two atoms in $P$ and $Q$ distributions; $H(\boldsymbol{\pi}) = -\sum_{i,k} \pi_{ik} \ln \pi_{ik}$ is the entropic regularization term. The transportation plan $\boldsymbol{\pi}$ can be interpreted as the topic proportions of documents in the corpus which are used as document representations in downstream tasks.

We now distinguish our proposed model with existing Wasserstein dictionary learning (WDL) models in [22, 23, 26] which also learn topics $\boldsymbol{B}$ and document topic proportions $\Lambda$. First, our model represents the collection of topics as a proper distributions $Q$, i.e. weights of topics sum to one while there is no constraint on the collection of topics in WDL models. Second, our model takes account of the word counts of documents, i.e., document length when we formalize the corpus empirical distribution $P$. In contrast, the normalized matrix to be decomposed in WDL does not carry information about the length of documents. Thanks to normalization constraints and word count consideration, our model shares a nice geometry interpretation with LDA as described in [27]. Moreover, this formulation also allows us to derive an efficient learning algorithm using alternating optimization, whereas, WDL learning procedure use gradient-based optimization which inherits slow convergence when the regularizer $\gamma$ is small.

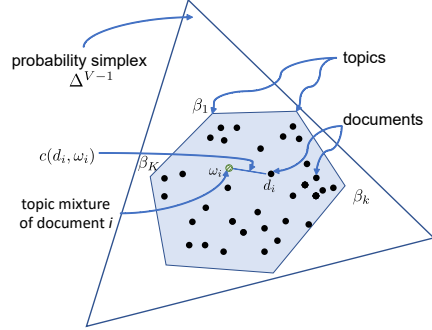

Figure 1: Geometric interpretation of LDA and OTLDA models. Vertices of the polytope represent the topics to be learned while black dots depicts (normalized) document observations. In LDA, the loss function is squared $l_2$ while we propose to use optimal transport distance in OTLDA.

**Geometric interpretation**. Let's recall the geometric surrogate loss function to the likelihood of LDA, $J_{GLDA} = \min_{\boldsymbol{\Omega}} \sum_{i=1}^{N} n_i \min_{\boldsymbol{\omega}_i \in \boldsymbol{\Omega}} \|\boldsymbol{\omega}_i - d_i\|_2^2$. We can intuitively depict this geometry in Figure 1. The topics form a polytope on the simplex of $\mathbb{R}^{V-1}$ which are learned from document observations using the squared $l_2$ distance $c(d_i, \omega_i)$ between a document $d_i$ and its topic mixture $\omega_i$. In our proposed framework, we instead use the optimal transport distance for $c(d_i, \omega_i)$ which allows us to incorporate the underlying geometry of word semantics via their distributional representation, e.g. word2vec.

## 3.2 Learning topics as a discrete optimal transport optimization

Given the corpus of $N$ documents in normalized bag-of-words format, and the number of topics $K$, we would like to learn the topic distributions $\mathbf{B} = \{\beta_k\}_{k=1}^K$, the topic proportions $\mathbf{b} = \{b_k\}_{k=1}^K$, and the topic proportions of documents $\boldsymbol{\pi} = \{\pi_{ik}\}_{i=1,k=1}^{N,K}$ which are the solution of the following optimization

$$\operatorname*{argmin}_{B,\mathbf{b},\boldsymbol{\pi}} \sum_{k=1}^K \sum_{i=1}^N \pi_{ik} \mathrm{OT}_\gamma(\beta_k, d_i) - \gamma H(\boldsymbol{\pi}).$$

We can solve this problem by using the principle of alternating optimization. For each iteration, we first fix the topic distributions $\mathbf{B}$ to compute the transportation plan from documents to topics and topic proportions. Then, we learn the new topic distributions $\mathbf{B}$ based on updated $\boldsymbol{\pi}$ and $\mathbf{b}$. However, before discussing how to update model parameters, we examine the way to compute Sinkhorn distance between document and topic $\mathrm{OT}_\gamma$ which is essential to our solution.

**Computing $\mathrm{OT}_\gamma(\beta_k, d_i)$.** Regularized optimal transport distance between document $i$ and topic $k$ is similar to word mover distance in [13] with an additional entropic regularization

$$\mathrm{OT}_\gamma(\beta_k, d_i) = \min_{\boldsymbol{\tau}} \sum_{u=1,v=1}^{V,V} \tau_{uv} d_{uv} - \gamma_\tau H(\boldsymbol{\tau}), \tag{3}$$

subject to $\sum_u \tau_{uv} = \omega_v$ and $\sum_v \tau_{uv} = \frac{n_{iv}}{n_i}$. Here, $H(\boldsymbol{\tau}) = -\sum_{uv} \tau_{uv} \ln \tau_{uv}$ and $d_{uv}$ is the distance between word $u$ and $v$ in the embedding space. In our framework, the distance can be any valid divergence such as *cosine* dissimilarity or *Euclidean* distance. We can use the Sinkhorn algorithm [5, 2] to efficiently compute the above distance.

**Fixing B, computing $\boldsymbol{\pi}$ and b**. We provide the result for updating $\boldsymbol{\pi}$ and $\mathbf{b}$ while fixing $B$ and defer the derivation to Supplementary material. Suppose that we have obtained $\{\beta_k\}_{k=1}^K$, our goal is to find optimal $\boldsymbol{\pi}_{jk}$ and $b_k$ for all $j, k$, which is the optimizer of

$$\min_{\mathbf{b},\boldsymbol{\pi}} \sum_{k=1}^K \sum_{i=1}^N \pi_{ik} c_{ik} - \gamma H(\boldsymbol{\pi}), \tag{4}$$

such that $\sum_k \pi_{ik} = \frac{n_i}{\sum_i n_i}$. Here, $c_{ik} = \mathrm{OT}_\gamma(\beta_k, d_i)$ is computed using Sinkhorn algorithm as described in Eq. (3). The transportation plan $\boldsymbol{\pi}$ solution of the problem in Eq. (4) is obtained at

$$\pi_{ik} = \frac{n_i}{\sum_i n_i} \frac{\exp(-\gamma c_{ik})}{\sum_v \exp(-\gamma c_{ik})},$$

and the proportion of topic $k$ is simply as $b_k = \sum_i \pi_{ik}$.

**Fixing $\boldsymbol{\pi}$ and b, computing B**. Assuming that document topic proportions $\boldsymbol{\pi}$ and the proportions of topics $\mathbf{b}$ are fixed, our goal is now to update $\beta_k$'s as the solution of the following optimization problem

$$\beta_k = \operatorname*{argmin}_{\beta} \sum_i \frac{\pi_{ik}}{b_k} \mathrm{OT}(\beta, d_i) \tag{5}$$

which turns out that $\beta_k$ is the (regularized) optimal transport barycenter which can be attained using Sinkhorn-based algorithms in [6, 2].

In summary, We summarize the learning procedure for our proposed framework in Algorithm 1. We have the following result guaranteeing the local convergence of this algorithm. We deferred detailed derivation of this algorithm and the full proof to the Supplementary Material.

**Proposition 1** *Algorithm 1 monotonically decreases the objective function of OTLDA* (2) *until local convergence.*

*Proof.* we sketch the proof as follows: updating on document topic proportions $\boldsymbol{\pi}$ and topic proportion $\mathbf{b}$ decreases the partial objective function in Eq. (4) and then updating topics $B$ decreases further the main objective function by optimizing the partial function in Eq. (5). Hence, each iteration will reduce the objective function in Eq. (2) until convergence.

**Algorithm 1** Optimal Transport based LDA (OTLDA)

---

**Require:** Data $D = \{d_i\}_{i=1}^n$; the number of topics $K$; the regularization hyper-parameter $\lambda > 0$.

**Ensure:** Learned topic proportions $\mathbf{b}$, topic distributions $\mathbf{B} = \{\beta_k\}_{k=1}^K$, and document topics proportions $\boldsymbol{\pi}$

Initialize topic proportions $\mathbf{b}$ and topic distributions $\{\beta_k\}_{k=1}^K$.

**while** not converged **do**

    1. Update document-topic propotions $\boldsymbol{\pi}$ and topic proportions $\mathbf{b}$:

    **for** $i = 1$ **to** $N$ **do**

        **for** $k = 1$ **to** $K$ **do**

            Compute document-topic propotion $\pi_{ij}$ as $\frac{n_i}{\sum_i n_i} \frac{\exp(-\lambda c_{ik})}{\sum_v \exp(-\lambda c_{ik})}$ where $c_{ik} = \mathrm{OT}_\gamma(\beta_k, d_i)$

            Update topic proportion $b_k = \sum_{i=1}^n \pi_{ik}$.

        **end for**

    **end for**

    2. Update topics $\beta_k$:

    **for** $k = 1$ **to** $K$ **do**

        Update topics $\beta_k$ as solution of barycenter $\underset{\beta}{\arg\min} \sum_i \frac{\pi_{ik}}{b_k} \mathrm{OT}(\beta, d_i)$.

    **end for**

**end while**

---

Table 1: Details of experimental copora. *#docs*: number of documents; *#words/doc*: the average number of words per document; *#vocabulary* : the number of word tokens.

| Dataset | #docs (N) | #vocabulary (V) | #words/doc |
|---------|-----------|-----------------|------------|
| 20NG | $18,845$ | $3,072$ | 80.8 |
| Wikipedia | $100,000$ | $4,962$ | 81.8 |
| 20NGshort | $3,197$ | $4,157$ | 9.8 |
| Twitter | $2,471$ | $4,359$ | 8.0 |

## 4 Experiments

To illustrate the performance of our proposed model, we conducted comprehensive experiments on several benchmark text datasets to fully evaluate the performance of OTLDA against the state-of-the-art topic models with and without word embeddings.

### 4.1 Experimental Setup

**Datasets**: our experiments are conducted on two categories of textual data including regular and short documents to demonstrate the robustness of OTLDA in terms of learning topic representation. For regular documents, we use two popular corpora including *20Newsgroups (20NG)* and *Wikipedia.*

- The *20Newsgroups* corpus consists of newsgroups post including approximately $18,000$ documents. We follow the pre-processing process in [8] in which the vocabulary is removed stop words, words with document frequency less than 100 times. Documents with less than one word are further removed from the corpus. We then use $80\%$ for training, $10\%$ for both validation and testing.

- The larger *Wikipedia* corpus is downloaded from wikipedia.com[1] including about 1.1 million documents. We also follow the pre-processing process in [12] We pre-process data using a vocabulary list taken from the top $10,000$ words in Project Gutenberg[2] and remove all words less than three characters. We extracted a subset of $100,000$ documents for our experiments.

We also use two short text corpora namely *20NGshort* and *Twitter* to demonstrate the strength of our model in the capability of modeling short texts. The *20NGshort* is a subset of documents from the *20NG* dataset with document length less than twenty-one and more than three words. We also

Table 2: Intrinsic and extrinsic topic coherence measures on four datasets: *20NG, 20NGshort, Tweets, Wikipedia.* Each row group depicts the performance of each dataset. Two models, LFTM and WNMF, took a very long time to run for large scale datasets like Wikipedia, hence $*$ denotes not available. Higher topic coherence means the model performs better.

| corpus | #topics | metrics | **EKmeans** | **LDA** | **ETM** | **LFTM** | **WNMF** | **OTLDA** |
|--------|---------|---------|-------------|---------|---------|----------|----------|-----------|
| 20NG | 50 | TC-UCI | -2.039 | -2.039 | -1.117 | -1.272 | -2.349 | **-0.988** |
|  |  | TC-UMass | 0.131 | 0.164 | 0.185 | -0.044 | 0.194 | **0.319** |
| 20NGshort | 20 | TC-UCI | -3.093 | -1.633 | -0.278 | -0.635 | -0.485 | **0.465** |
|  |  | TC-UMass | -0.982 | -0.431 | **-0.355** | -0.984 | -0.696 | -0.388 |
| Tweets | 20 | TC-UCI | -2.98 | -2.66 | -1.149 | -1.61 | -0.672 | **0.162** |
|  |  | TC-UMass | -0.989 | -0.516 | **-0.354** | -0.984 | -0.623 | -0.468 |
| Wikipedia | 100 | TC-UCI | -1.783 | -1.086 | -0.529 | $*$ | $*$ | **0.57** |
|  |  | TC-UMass | 0.116 | 0.219 | 0.231 | $*$ | $*$ | **0.256** |

use 2011 and 2012 micro-blog tracks at Text REtrieval Conference (TREC)[3] which includes $2,471$ tweets of the average length of $8$. We summarize the statistics of datasets in Table 1.

**Baseline methods and settings**: we evaluate the proposed method in both quantitative and qualitative aspects. For the quantitative aspect, we use topic coherence (TC) to measure the intrinsic performance of the methods while document classification (DCS) is used as the extrinsic evaluation metric. We evaluate the performance of OTLDA against existing topic model methods[4] including naive model of clustering word embedding vectors using k-means (*EKmeans*); notable Latent Dirichlet Allocation (*LDA*) [3] - topic modeling based on word co-occurrences; two extensions of LDA incorporating word embedding information: *ETM* [8] - a neural topic model; and *LFTM*[5] [19] - probabilistic topic model with embedding latent feature; and WNMF [22] a Wasserstein distance-based topic model. We used the default parameters given with the source code or the best settings reported in the papers for the ETM, *LFTM, WNMF* models while for LDA, we use the default parameter given by the Gensim package. For all models with embedding, we use the pre-trained *word2vec* from Google.[6]

**Evaluation metrics**: we use *topic coherence* and *F-score and accuracy* of document classification to evaluate the performance of our proposed model. Topic coherence is a quantitative measure to evaluate the interpretability of the learned topics which is highly correlated to human judgment [16]. In this paper, we use two versions of topic coherence, extrinsic UCI measure (TC-UCI)[7] [18, 21] and intrinsic UMass measure (TC-UMass) [16]. Both measures share the same high-level idea, however, the TC-UCI uses the word correlation statistics from a universal corpus, e.g. Wikipedia, while the UMass uses the training corpus as the reference corpus.

## 4.2 Experimental Results

**Quantitative results**: we ran our proposed methods with several regularization terms including $0.05,\ 0.1,\ 1\ 50$ and choose the best performance among them. We found that with regular documents, large regularizer $\lambda$, e.g $50$, provide a better topic coherence while a smaller regularizer, e..g. $\lambda = 0.05$, is more suitable for short documents. We also report the effect of regularizer on the topic coherence on the corpus in the Supplementary Material. Table 2 reports the comprehensive topic coherence measures on four corpora. In the table, the best and the second-best score of each corpus are boldface and underline highlighted respectively. We observe that when dealing with regular documents, the OTLDA outperforms baseline methods in terms of both intrinsic and extrinsic topic coherence, meanwhile, ETM and WNMF are the second-best of TC-UCI and TC-UMass re-

Table 4: Comparison of cherry-picked top three ETM and OTLDA topics on Wikipedia corpus related to *government* keyword.

| ETM | government, foreign, department, minister, public, ministry, development, office, economic, council, service |
| | pay, money, financial, million, cost, economic, rate, tax, funds, government, increase |
| | political, movement, government, independence, party, freedom, war, social, president, union, republic |
| OTLDA | community, provide, business, educational, develop, government, **help**, specific, development, **encourage**, work |
| | government, legislature, state, legislative, political, commission, judicial, senate, committee, governor, national |
| | government, community, education, educational, **public**, business, local, **university**, national, organization, **private** |

spectively. For short text, OTLDA performs much better in comparison with its baselines in terms of extrinsic coherence, however, it is a bit behind the intrinsic performance of ETM. Note that the TC-UMass values of all models are small since the length of documents is small which leads to low empirical word co-occurrence.

To compare the extrinsic predictive performance, we use document classification as a downstream task. After topics are learned from the corpus, we use trained models to predict the document-topic proportions for testing and validation subsets. We use the learned document-topic proportions of training data and its labels to train a Support Vector Machine classifier (SVC). We use *SVC* model in *scikit-learn* with the default parameters to train and report the classification performance in Table 3. We choose the three best models in topic coherence to compare with our proposed model. Table 3 depict the results of document classification. OTLDA surpass the baseline methods in terms of both accuracy and F-score.

We can intuitively justify the better semantic coherence of learned topics using underlying mechanism of word embeddings. The word embeddings are resulted from learning co-occurring of words in documents, therefore when two words $w_u$ and $w_v$ appear more frequently in the corpus, their embeddings are more similar, i.e. $c_{uv}$ is small. Our proposed model aims to optimize Eq. (5) which will put a higher value on $\gamma_{uv}^i$ . As a consequence, the pair of $w_u$ and $w_v$ usually gets higher weight in topic $\beta_k$. When computing topic coherence, we usually choose top words with high weights, it is more like this pair of words to present in the list of the top words which may produce bigger the numerator of coherence formula. Another property that our proposed model process is the clustering characteristic which means closer documents in terms of optimal transport (aka word mover distance - WMD) will have similar topic proportion vectors. We also knew that WMD provides good distance for documents in text classification in [13]. Our results in the downstream classification task are orthogonal with their results.

**Qualitative results**: we cherry-pick three topics related to *government* (the keyword in the top ten words of the topic) learned by ETM and OTLDA models to investigate the qualitative performance of the models. The top row of Table 4 depicts topics from ETM models while the bottom row is the topics from OTLDA models. The ETM topics are relatively too broad and general, and the second topic is not much related to *government*. The topics learned by OTLDA are more detailed. For example, both the first and third topics mention *community education (of) government and business.* But the first is about *help and encourage* while the other topic is about *public and private university (ies).* The results reflect that OTLDA prefers to learn topics with fine granularity.

Table 3: Classification performance of 20NG corpus using document-topic proportions as vector representation for documents.

| metrics | ETM | WNMF | OTLDA |
|---|---|---|---|
| Accuracy | 0.469 | 0.412 | **0.478** |
| Precision | **0.477** | 0.437 | 0.471 |
| Recall | 0.455 | 0.423 | **0.464** |
| F1 | 0.443 | 0.421 | **0.459** |

## 5   Conclusion

We have presented an optimal transport framework called OTLDA for topic modeling which is grounded on the convex geometric perspective of topics in relation to document observation. Based on this interpretation, we have developed an efficient learning algorithm using alternating optimization. Extensive experiments illustrate that the proposed model outperforms competitive approaches

in terms of topic coherence as well as better classification accuracy for downstream tasks. We also wish to highlight the conceptual difference between OTLDA and Wasserstein dictionary learning, WNMF. OTLDA takes the number of word counts of documents into account as well as using normalized learned topic proportions, while WNMF does not. Our approach to topic modeling may lend itself naturally to other LDA extensions due to its simplicity and flexibility. Future direction can, for example, consider dynamic or supervised settings.

## Broader Impact

Topic modeling is one of the most popular tools for text understanding and text mining. Understanding large document collections, retrieving scientific findings related to some disease, or discovering the trends of current news of the election are some examples of domain application of topic models. Our work provides a new theoretical framework to improve the interpretation of learned topics. Hence, it will be beneficial to some current applications that use directly from learned topics of the topic models. Learning more interpretable topics from a huge corpus can also be important for decision making support applications. As the work addresses a fundamental research problem we believe that it does not put anyone at disadvantages. Our method is data-driven which completely depends on the input corpora. The potential of unfairness may come from the process of data collection.

## Footnotes

[1] We used the dump of the English Wikipedia on June 02, 2015.

[2] https://en.wiktionary.org/wiki/Wiktionary:Frequency_lists/PG/2006/04/1-10000

[3]This is originally from https://trec.nist.gov/data/microblog.html and pre-processed by Jipeng et. al. at https://github.com/qiang2100/STTM

[4]We are unable to compare with DWL [26] since the code of the model is not publicly available.

[5]Authors develop two class of models for regular (LFLDA) and short (LFDMM) text. Depending on the dataset, we choose the corresponding method.

[6]https://code.google.com/archive/p/word2vec/

[7]We used the Palmetto package (http://palmetto.aksw.org) with Wikipedia corpus.

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
