[Supplementary Material]

# Supplementary material for "OTLDA: A Geometry-Aware Optimal Transport Approach for Topic Modeling"

## 1 Derivations for OTLDA learning

In this section, we present our derivation for learning OTLDA models. Let us recall the objective function of OTLDA as follows

$$J\left(\boldsymbol{B}, \mathbf{b}, \boldsymbol{\pi}\right) = \min_{\boldsymbol{B}, \mathbf{b}, \boldsymbol{\pi}} \sum_{k=1}^{K} \sum_{i=1}^{N} \pi_{ik} \mathrm{OT}_{\gamma}(\beta_k, d_i) - \gamma H\left(\boldsymbol{\pi}\right), \tag{1}$$

such that $\sum_k \pi_{ik} = \frac{n_i}{\sum_i n_i}$, $\sum_i \pi_{ik} = b_k$. If we fix $\boldsymbol{B}$, the objective function now reads

$$J_{\pi}\left(\mathbf{b}, \boldsymbol{\pi}\right) = \min_{\mathbf{b}, \boldsymbol{\pi}} \sum_{k=1}^{K} \sum_{i=1}^{N} \pi_{ik} c_{ik} - \gamma H\left(\boldsymbol{\pi}\right), \tag{2}$$

such that $\sum_k \pi_{ik} = \frac{n_i}{\sum_i n_i}$, $\sum_i \pi_{ik} = b_k$. Here we denote $c_{ik} = \mathrm{OT}_{\gamma}(\beta_k, d_i)$ and $H\left(\boldsymbol{\pi}\right) = -\pi_{ik} \ln \pi_{ik}$. Using the Lagrangian multiplier for the constraints, we need to optimize the objective function

$$J_{\pi}\left(\mathbf{b}, \boldsymbol{\pi}\right) = \min_{\mathbf{b}, \boldsymbol{\pi}} \sum_{k=1}^{K} \sum_{i=1}^{N} \pi_{ik} c_{ik} - \gamma H\left(\boldsymbol{\pi}\right) + \lambda_1 \sum_i \left(\sum_k \pi_{ik} - \frac{n_i}{\sum_i n_i}\right) + \lambda_2 \sum_i \left(\sum_i \pi_{ik} - b_k\right), \tag{3}$$

Taking derivatives of Eq. (3) with respect to $\pi_{ik}$ and setting to zero, we have

$$\frac{\partial J}{\partial \pi_{ik}} = c_{ik} + \gamma \ln \pi_{ik} + \gamma + \lambda_1 + \lambda_2 = 0,$$

which means $\pi_{ik} = \exp\left(-c_{ik}/\gamma\right) \exp\left(-\left(\gamma + \lambda_1 + \lambda_2\right)\gamma\right)$. Considering the condition $\sum_k \pi_{ik} = \frac{n_i}{\sum_i n_i}$, we can compute

$$\frac{n_i}{\sum_i n_i} = \exp\left(-\left(\gamma + \lambda_1 + \lambda_2\right)\gamma\right) \sum_i \exp\left(-c_{ik}/\gamma\right)$$

$$\exp\left(-\left(\gamma + \lambda_1 + \lambda_2\right)\gamma\right) = \frac{n_i}{\sum_i n_i} \frac{1}{\sum_i \exp\left(-c_{ik}/\gamma\right)}$$

Hence, we obtain

$$\pi_{ik} = \frac{n_i}{\sum_i n_i} \frac{\exp\left(-c_{ik}/\gamma\right)}{\sum_i \exp\left(-c_{ik}/\gamma\right)}. \tag{4}$$

12   The value of $b_k$ is computed using the constraint $b_k = \sum_i \pi_{ik}$.

13   We also have the update for $\boldsymbol{B}$ from this (sub-)objective

$$J\left(\boldsymbol{B}\right) = \sum_{k=1}^{K} \min_{\beta_k} \sum_{i=1}^{N} \frac{\pi_{ik}}{b_k} \mathrm{OT}_\gamma(\beta_k, d_i). \qquad (5)$$

14   The inner optimization associated with each $\beta_k$ is an optimal transport barycenter itself

$$\beta_k = \underset{\beta}{\mathrm{argmin}} \sum_{i=1}^{N} \frac{\pi_{ik}}{b_k} \mathrm{OT}_\gamma(\beta, d_i),$$

15   which can be used the Sinkhorn-based barycenter algorithms [1] to solve.

## 16   2   Proof of proposition 1

17   We restate the Proposition 1 as follows

18   **Proposition 1** *Algorithm 1 monotonically decreases the objective function of OTLDA* (1) *until local*
19   *convergence.*

20   *Proof.* Given the formulation of Algorithm 1, we would like to demonstrate its convergence to a local
21   solution of objective function Eq. (1) in Proposition 1. We denote $\boldsymbol{B}^{(t)}, \mathbf{b}^{(t)}, \boldsymbol{\pi}^{(t)}$ as the update of
22   topics, topic weights, document topic proportions in step $t$ of Algorithm 1 for $t \geq 0$. Additionally,
23   let $c_{ik}^{(t)}$ be the cost value between document $i$ and topic $k$ at step $t$, i.e., $c_{ik}^{(t)} = \mathrm{OT}_\gamma(\beta_k^{(t)}, d_i)$ for all
24   $i, k$. We also denote $S_{\boldsymbol{\pi}} \triangleq \left\{ \boldsymbol{\pi} : \sum_{k=1}^{K} \pi_{ik} = \frac{n_i}{\sum_i n_i} \; \forall 1 \leq i \leq n \right\}$. Furthermore, we denote

$$\mathcal{L}\left(\mathbf{b}, \boldsymbol{B}\right) \triangleq \min_{\boldsymbol{\pi} \in \Pi(\bar{\boldsymbol{n}}, \mathbf{b})} \sum_{k=1}^{K} \sum_{i=1}^{N} \pi_{ik} \mathrm{OT}_\gamma(\beta_k, d_i) - \gamma H\left(\boldsymbol{\pi}\right),$$

25   where $\bar{\boldsymbol{n}}$ is a vector of $N$ dimension of document normalized word counts, i.e. $\bar{n}_i = \frac{n_i}{\sum_i n_i}$ ; and
26   $\Pi\left(\bar{\boldsymbol{n}}, \mathbf{b}\right) \triangleq \left\{ \boldsymbol{\pi} \in \mathbb{R}^{N \times K} \mid \pi \mathbf{1}_N = \bar{\boldsymbol{n}}, \; \pi^{\mathsf{T}} \mathbf{1}_K = \mathbf{b} \right\}$ is the set of transportation plans between $\bar{\boldsymbol{n}}$ and
27   $\mathbf{b}$.

28   For any $t \geq 0$, it is clear that

$$\mathcal{L}\left(\mathbf{b}^{(t)}, \boldsymbol{B}^{(t)}\right) = \min_{\boldsymbol{\pi} \in \Pi(\bar{\boldsymbol{n}}, \mathbf{b})} \sum_{k=1}^{K} \sum_{i=1}^{N} \pi_{ik} c_{ik}^{(t)} - \gamma H\left(\boldsymbol{\pi}\right)$$

$$\geq \min_{\boldsymbol{\pi} \in S_{\pi}} \sum_{k=1}^{K} \sum_{i=1}^{N} \pi_{ik} c_{ik}^{(t)} - \gamma H\left(\boldsymbol{\pi}\right) \qquad (6)$$

$$\geq \sum_{k=1}^{K} \sum_{i=1}^{N} \pi_{ik}^{(t+1)} c_{ik}^{(t)} - \gamma H\left(\boldsymbol{\pi}^{(t+1)}\right)$$

29   We can obtain the first inequality due to $\Pi\left(\bar{\boldsymbol{n}}, \mathbf{b}\right) \subset S_{\pi}$ while the last inequality due to Eq. (4)
30   which is the minimizer we obtained. From the update of topics $\boldsymbol{B}$ in Eq. (5), we have

$$\sum_{k=1}^{K} \sum_{i=1}^{N} \pi_{ik}^{(t+1)} c_{ik}^{(t)} - \gamma H\left(\boldsymbol{\pi}^{(t+1)}\right) = \sum_{k=1}^{K} \sum_{i=1}^{N} \pi_{ik}^{(t+1)} \mathrm{OT}_\gamma(\beta_k^{(t)}, d_i) - \gamma H\left(\boldsymbol{\pi}^{(t+1)}\right)$$

$$\geq \sum_{k=1}^{K} \sum_{i=1}^{N} \pi_{ik}^{(t+1)} \mathrm{OT}_\gamma(\beta_k^{(t+1)}, d_i) - \gamma H\left(\boldsymbol{\pi}^{(t+1)}\right) \qquad (7)$$

$$\geq \min_{\boldsymbol{\pi} \in \Pi(\bar{\boldsymbol{n}}, \mathbf{b})} \sum_{k=1}^{K} \sum_{i=1}^{N} \pi_{ik} c^{(t+1)} - \gamma H\left(\boldsymbol{\pi}\right)$$

$$= \mathcal{L}\left(\mathbf{b}^{(t+1)}, \boldsymbol{B}^{(t+1)}\right).$$

31 Combining the results from Eqs. (6) and (7), for any $t \geq 0$, the following holds

$$\mathcal{L}\left(\mathbf{b}^{(t)}, \boldsymbol{B}^{(t)}\right) \geq \mathcal{L}\left(\mathbf{b}^{(t+1)}, \boldsymbol{B}^{(t+1)}\right).$$

32 As a consequence, we achieve the conclusion of Proposition 1.

## 3 Additional experimental results

34 In our proposed model, the regularizer parameters $\lambda$s play as smoothing factors for learning topics.
35 The topic coherence of learned topics is affected by these parameters. We experimented with the
36 effects of those parameters on topic coherence on two datasets: *20Newsgroups* and *20NGshort*.
37 Figure 1a depicts the UCI topic coherence obtained with different values of $\lambda$ for *20Newsgroups*
38 dataset. The range for good topic coherence with this corpus is from 10 to 30. Figure 1b show the
39 effect of $\lambda$ to topic coherence which infers that the coherence gets its good value with small $\lambda$.

(a) *20Newsgroups* dataset        (b) *20NGshort* dataset

Figure 1: UCI Topic coherence of learned topics with *20Newsgroups* and *20NGshort* datasets with respect to entropic regularizer parameters. We set all three regularizer parameters $\lambda$s equally.

40 We report the top ten words in top topics which have the highest UCI topic coherence values of two
41 models, EMT and OTLDA learned with two datasets 20NG and Tweets. Table 1 depicts top ten
42 topic among 50 learned topics while Table 2 shows top five topics.

## References

44 [1] Marco Cuturi and Arnaud Doucet. Fast computation of Wasserstein barycenters. 2014.

Table 1: List of top ten words of top ten topics with highest UCI topic coherence learned by ETM and OTLDA with *20NG* dataset

| | No. | Top ten words |
|---|---|---|
| ETM | 1 | god, people, jesus, christian, bible, christians, church, religion, christ |
| | 2 | food, medical, science, blood, disease, doctor, medicine, pain, treatment |
| | 3 | mail, access, software, net, information, info, computer, phone, fax |
| | 4 | ftp, information, mail, anonymous, send, pub, internet, list, email |
| | 5 | drive, scsi, card, windows, dos, disk, pc, system, mac |
| | 6 | good, time, writes, article, make, back, lot, put, thing |
| | 7 | power, water, ground, current, circuit, wire, good, high, cover |
| | 8 | people, make, good, time, writes, things, article, thing, put |
| | 9 | president, mr, clinton, government, people, health, secretary, jobs, program |
| | 10 | space, nasa, gov, earth, toronto, henry, moon, data, launch |
| OTLDA | 1 | god, church, jesus, faith, christ, christian, bible, moral, atheists, morality, lord |
| | 2 | info, version, ibm, software, email, unix, send, user, internet, machines, computer |
| | 3 | card, drives, monitor, mac, scsi, drive, dos, ram, disk, controller, installed |
| | 4 | sense, person, understand, reason, fact, life, matter, claim, evidence, clear, people |
| | 5 | mike, kevin, tom, san, doug, cup, ca, roger, patrick, dave, michael |
| | 6 | country, crime, gun, death, police, attack, court, civil, happen, killed, brought |
| | 7 | turkish, armenian, armenians, turkey, soldiers, nazi, armenia, israel, israeli, peace, genocide |
| | 8 | find, mentioned, make, give, question, read, great, called, time, real, good |
| | 9 | nasa, space, moon, henry, earth, launch, surface, toronto, flight, ames, spencer |
| | 10 | country, people, fact, happen, today, matter, talk, freedom, responsible, sense, bring |

Table 2: List of top ten words of top ten topics with highest UCI topic coherence learned by ETM and OTLDA with *Tweets* dataset

| | No. | Top ten words |
|---|---|---|
| ETM | 1 | judge, health, law, debt, care, state, rule, immigration, federal, aid, reform |
| | 2 | open, stock, market, help, job, start, share, free, set, financial, find |
| | 3 | fishing, fish, fly, ice, bass, aquarium, tip, trout, bait, caught, big |
| | 4 | king, speech, award, oscar, nomination, win, top, best, film, academy, lead |
| | 5 | medium, light, sheen, charlie, party, red, white, stripe, tea, well, fat |
| OTLDA | 1 | law, judge, court, constitutional, government, judicial, ruling, rule, federal, constitution, judiciary |
| | 2 | fishing, fish, trout, walleye, fisherman, fished, angler, largemouth, catfish, shad, marlin |
| | 3 | college, scholarship, education, school, university, academic, academy, student, undergraduate, social, award |
| | 4 | debt, financial, money, pay, investment, income, tax, buy, budget, help, plan |
| | 5 | diet, weight, nutrition, acai, nutritional, healthy, health, nutrient, berry, calorie, vitamin |