[Reviews · NeurIPS 2020]

Review 1

Summary and Contributions: This paper proposes a document-topic modeling formulation based on a Optimal Transport distances, showing improvements in topic coherence and a downstream document classification task.

Strengths: The modeling approach, while somewhat intricate, seems internally coherent. The framing and theoretical connections are interesting, and the experimental results seem strong.

Weaknesses: The coherence results in Table 2 seem consistently and significantly better than strong baselines, but I have no idea why that should be the case. What about this formulation leads to topics better semantic coherence, or utility in downstream classification tasks in Table 3? Without some at least hypothesized mechanism here I am kind of suspicious of the results. Overall I find the entire approach to lack clear motivations beyond the aesthetic appeal of the geometric interpretation and OT distance machinery, which admittedly seems nice. I was left with open questions around which kinds of problems we expect this approach to be superior on or what kinds of extensions could be more naturally expressed in this framework. Why use this somewhat involved machinery instead of some other approach?

Correctness: L269: tuning the regularizer parameter lambda for the proposed method without also doing so for the other approaches does not seem correct. I don't find the qualitative results very compelling. How does this approach outperform on coherence versus baselines that leverage the pretrained semantic embeddings? This seems surprising and I would be interested to see this explained more thoroughly.

Clarity: I found L137-182 to be fairly dense, especially regarding the "nested" OT problem being solved. The language is clear and precise, and the figure is helpful, but some higher-level setup or motivation might make the flows easier to follow. Given all this, the optimization procedure on the subsequent two pages was clear enough.

Relation to Prior Work: L157-168: what is the benefit or advantage to these two differences vs WDL? I understand these are differences, but the associated practical or theoretical gains were not clear to me. Other than the unfortunate omission of DWL, the experimental baselines seem strong.

Reproducibility: Yes

Additional Feedback: The paper could be strengthened with some more explicit motivation or justification for why the particular OT formulation can be expected to yield improved results, what additional benefits or flexibility we should expect from this formulation, or why this formulation could have a significant research impact. L103: it was not immediately obvious to me whether Big Omega was the set of topic vertices only, or their convex combinations as well. L184: what is the norm here? Figure 1 was very helpful. It seems like one could use different gamma entropic regularization parameters for the "inner" (topic-document) and "outer" (document-topic) OT problems, was this explored? Table 3: caption has a typo 2NG -> 20NG Regarding results quality and convergence efficiency, it would have very helpful to see experimental comparisons against WDL. There is nothing that these authors can do about it, but it is unfortunate that prior NeurIPS publication "Distilled Wasserstein Learning" lacks available source code for comparison. That said, was there any reason not to compare against other Wasserstein-based approaches? Are learning times/complexity for this model dramatically better (worse?) or comparable to the baselines? I have read and considered the author feedback.


Review 2

Summary and Contributions: The paper proposes a method for encoding geometric information such as from word embeddings into topic model training, using an optimal transport formulation. Although there has been recent work with related ideas, the approach here respects more of the semantic constraints of models like LDA (e.g. normalized distributions) and is found to have better performance and faster training.

Strengths: Optimal transport approaches to topic modeling are a relatively new and promising direction. The paper improves over the state of the art in this space. Incorporating word embedding information into topic modelings has long been an important goal, and although substantial progress has been made on this before, using embeddings to encode semantics into the geometry of the learning problem is an elegant and interesting strategy that could lead to follow-on research. The model and learning algorithm are reasonable and sound. Local convergence is established for the learning algorithm. The experimental setups are mostly reasonable and the results are positive.

Weaknesses: Although optimal transport approaches to topic modeling are relatively new, this is not the first work to attempt this, which reduces the novelty somewhat. The claimed advantages over e.g. [26] are mainly encoding normalization constraints and word count information, and a more efficient optimization algorithm. These are somewhat incremental improvements, especially by the standards of the NeurIPS conference, although they do lead to better performance. Only 20-100 topics were used in the experiments, which is rather low by today's standards. The classification results to evaluate downstream extrinsic performance, while appreciated, are rather limited. Only one dataset is used in the experiment. The results for all of the compared methods appear very poor, and only 3 of the models were considered. It is important to compare to baseline methods like tf-idf features.

Correctness: The claims are mainly supported. The technical approach seems correct, although I do not have the expertise to check all of the details. One concern is that in the experiments, default or best-reported hyper-parameter values were used for the baselines, while the proposed method used a (small) grid search on its regularization parameter lambda. This gives the proposed method a slight advantage. It was unclear whether the best lambda value was chosen based on performance on a validation set or on the test set.

Clarity: The paper is mainly well written and well argued. However, I found the notation very confusing in Section 3 which hampered understanding the work. Sometimes symbols are used before they are properly defined. There is a general lack of textual explanation to provide intuitive understanding for the mathematical notation. The dimensionality of vectors and matrices are generally not specified. Most notational symbols are non-standard for topic models such as LDA. It would help to include a notation table for the reader to refer back to. Along the same lines, P and Q are each referred to as a distribution, but they seem to encode a collection of distributions. The semantics of these variables and their notation need to be clarified.

Relation to Prior Work: Section 2.2 discusses previous approaches based on word embedding semantics and deep neural networks, but does not clearly explain the limitations of those approaches that are addressed by the present work. The paper includes a discussion of its relation to related optimal transport approaches in Section 3.1, but it could make it clearer in the abstract and introduction that it is not the first paper to develop such an approach, and what its precise novelty is relative to those papers.

Reproducibility: Yes

Additional Feedback: Some minor suggestions and typos: -Line 21, missing an "and" -Line 33, "while other developed" - "while other authors developed" -Line 50, and elsewhere in the paper, it is stated that LDA/PLSI use a squared Euclidean loss/distance. This is untrue - both models use likelihood based inference with a multinomial model, and/or Bayesian inference. The older LSI model uses a squared loss, but even the PLSI paper argued that this is insufficient (the implicit Gaussian assumption from squared errors does not hold with small counts as in text data), which motivates the probabilistic modeling approach in PLSI and LDA. -Line 81, I believe that one of the sums over v should be a sum over u. -Line 87, punctuation is missing before the mathematical notation in this sentence. -Line 104, "Authors" - "The authors" -Line 110, reference [14] is to a 2014 paper by Mikolov. The other papers by Mikolov by 2013 are more fundamental references which are better here, especially: Mikolov, T., Sutskever, I., Chen, K., Corrado, G. S., & Dean, J. (2013). Distributed representations of words and phrases and their compositionality. In Advances in Neural Information Processing Systems (pp. 3111-3119). -Line 123, "which neglects to word semantics" -Line 128, "with jointly learning cost function" - "by jointly learning a cost function" -Line 136, "bag-of-word" -Line 189, "number of topic" -Line 193, "alternative optimization" - should this be "alternating optimization"? This is repeated elsewhere in the paper. -Line 202, capitalize "euclidean" -Algorithm 1, "regularized" - "regularization" -Line 293, "depicts topics ETM" -Line 308, "classification accuracy downstream tasks" --- Comments after reading the rebuttal: My concern about hyper-parameter selection was addressed more or less adequately in the rebuttal. The method may still get an unfair advantage since it was tuned on the target dataset rather than another dataset from an earlier paper, but since only 4 values were considered the advantage would be slight. The other reviews didn't raise any other red flags. I'm inclined to leave my score as-is.


Review 3

Summary and Contributions: The paper presents a topic model based on optimal transport and word-movers' distance.

Strengths: Topic modeling continues to provide a useful test ground for inference methods, and this is no exception. I'm not particularly interested in the quantitative performance of topic evaluation metrics one way or another. For me the value is in demonstrating a potentially useful algorithmic approach and identifying where it is different from existing methods.

Weaknesses: 8M tokens is not large-scale. None of these collections is competitive in modern terms, particularly in the vocabulary -- I would want to see at least 50k distinct terms. This makes me very concerned about efficiency. I'd like to know wallclock times and scaling with corpus size and vocabulary dimension. The introduction mentions embeddings, but I actually can't tell whether the proposed method uses pre-trained embeddings or not. p4/180 is another cryptic remark: it could use word2vec, but does it? I don't see any mention in the rest of section 3. If it doesn't, this is a huge advantage over ETM and should be emphasized. If it does, the impact of the quality of the embedding space should be evaluated. The difference between count and proportion vector for WNMF doesn't seem that important, but the difference between being unable to train on an 8M token corpus (WNMF) and getting pretty good performance (OTLDA) seems massive. What accounts for this difference?

Correctness: Using a Gutenberg vocabulary for Wikipedia is a strange choice. Why would we expect out-of-copyright books to be a good source of vocab for a modern encyclopedia? The vocabulary size (4k) is extremely small for this size collection as a result. Gensim's default LDA implementation is known to work poorly or collections that are not extremely large. The qualitative results in table 4 are not particularly compelling. The goal in this type of evaluation is to complement the quantitative results to show that -.355 vs -.388 is something you would actually notice. I want to see a range of topic examples. I'm a bit concerned about distinctiveness between topics. (If you think "tax" and "money" is not related to government, I have bad news for you.) Usually the UCI metric is a PMI, which can be positive or negative, but the UMass metric is a log of a ratio, and can't be positive. I'd actually prefer using the same function, but just varying the reference corpus (internal vs. external) if that's what's happening, but it should be mentioned.

Clarity: Yes.

Relation to Prior Work: Yes.

Reproducibility: Yes

Additional Feedback:

[Author Response · NeurIPS 2020]

We thank all the reviewers for their time, valuable and encouraging feedback and recommendations for improvement.
In the following, we address their concerns and questions.

*Motivations, intuitions, and formulation lead to better semantic coherence and downstream tasks* (**R1, R2**): In short,
we can justify the better semantic coherence of learned topics using underlying mechanism of word embeddings and
optimizer in Eq. (5) which is $\beta_k = \arg\min_\beta \sum_i \frac{\pi_{ik}}{b_k} \sum_{u,v} \gamma^i_{uv} c_{uv}$. Our intuitive explanation is as follows. The word
embeddings are resulted from learning co-occurring of words in documents, therefore when two words $w_u$ and $w_v$
appear more frequently in the corpus, their embeddings are more similar, i.e. $c_{uv}$ is small. Our proposed model aims to
optimize Eq. (5) which will put a higher value on $\gamma^i_{uv}$. As a consequence, the pair of $w_u$ and $w_v$ usually gets higher
weight in topic $\beta_k$. When computing topic coherence, we usually choose top words with high weights, it is more
like this pair of words to present in the top words list which may produce bigger the numerator of coherence formula.
Another property that our proposed model process is clustering characteristic which means closer documents in terms
of optimal transport (aka word mover distance - WMD) will have similar topic proportion vectors. We also knew that
WMD provides good distance for documents in text classification in [1]. Our results in downstream classification task
are orthogonal with their results.

*Tuning the regularizer parameter lambda* (**R1,R2**): In fact, we did **not** heavily tune the regularizer parameter. We ran
the model with four settings of the regularizer parameter is to check the sensitivity of our model. On the contrary, for
the baseline approaches, we chose the best-reported values in their papers which we think had already gone through a
tuning procedure.

*The advantage of considering word counts in our proposed model* (**R1,R3**): When dealing with a varied document
length corpus, thank to word counts consideration, our model can up-weight longer documents while down-weighting
shorter ones. We have demonstrated that our model can handle short document datasets such as 20NGshort or Tweets
better. Word count weighting also provides the connection between our proposed model and LDA.

*Some minor suggestions, notations, and typos* (**R1,R2, R3**): We appreciate your pointing and constructive suggestions.
We will improve the manuscript with the suggestions.

**R1:** *Exploration of entropic regularization parameters*: There are different gamma entropic regularization parameters.
As we mentioned we did not tune these parameters when comparing to baseline approaches. We leave the investigation
of the effects the entropic regularization parameters as future work.

*Not to compare against other Wasserstein-based approaches*: There are two Wasserstein-based approaches which can
solve the problem namely WDL and DWL. Unfortunately, the code of DWL is not publicly available, we are not able to
compare with. One of our baselines is DWL which we called WNMF in the paper.

*Times/complexity for this model*: In comparison with WNMF, our model is much faster since using Sinkhorn-based
algorithms to learn while WNMF runs Sinkhorn-based algorithms (forward) then compute the gradient to update the
model (backward). In comparison with neural topic models like ETM, our model is slower since neural topic models use
amortized variational inference to learn. We did not include running time since the code are implemented in different
platforms or programming languages. For instance, WNMF is implemented with Matlab/C++, ETM is coded with
PyTorch while our model is implemented with plain python using POT library.

*L184*: It is $L_1$ norm. We will clarify it.

**R2:** *Only 20-100 topics were used*: Choosing the number of topics in topic models is a challenging task which is not
the main focus in our experiments. Our strategy for selecting the number of topics for each dataset follows existing
work in the literature. Moreover, in practice, learned topics are usually inspected by a human for the use of visualization
or understanding, it is impractical to deal with a very large number of topics.

*I found the notation very confusing*: We will clarify symbols and notations in the revised version. In particular, thanks
for your suggestion to include a table to summarize the notations, we will implement that.

**R3:** *Datasets are not large*: We agree that the datasets we used are not considered as modern datasets. In this paper, we
would like to demonstrate a novel tool to solve the topic modeling problem. Scaling up the current model to massive
datasets is one of our future work.

*Embeddings*: We did mention the use of word2vec embeddings in our experiments in lines 257–258.

*Qualitative results and UCI metric*: Given space restriction, we had to make a choice to balance the theory and
experimental results, we shall aim to improve the post-analysis results as well as more topics in the supplementary
materials

[1] Matt Kusner, Yu Sun, Nicholas Kolkin, and Kilian Weinberger. From word embeddings to document distances. In
*International conference on machine learning*, pages 957–966, 2015.


[Meta-Review · NeurIPS 2020]

Discussion of this paper tended positive, and reviewers suggested their opinion of the work improved after reading the rebuttal. They agree this paper is a timely contribution to the literature and that the experiments are compelling. In the camera ready, please address promises in the rebuttal and requests for edits/experiments in the reviews. In particular, please address concerns about motivation/intuition and regularizer tuning.